# The Position-Reputation-Information (PRI) scale of individual prestige

**Richard E. W. Berl** [ID] [1]*, **Alarna N. Samarasinghe**[2], **Fiona M. Jordan**[2,3], **Michael C. Gavin**[1,3]

**1** Department of Human Dimensions of Natural Resources, Colorado State University, Fort Collins, Colorado, United States of America, **2** Department of Anthropology and Archaeology, University of Bristol, Bristol, United Kingdom, **3** Max Planck Institute for the Science of Human History, Jena, Germany

* rewberl@colostate.edu

**Data Availability Statement:** All relevant data are available within the paper and its Supporting Information files, and are also accessible on the Open Science Framework "Cultural Transmission"

## Abstract

Prestige is a key concept across the social and behavioral sciences and has been implicated as an important driver in the processes governing human learning and behavior and the evolution of culture. However, existing scales of prestige fail to account for the full breadth of its potential determinants or focus only on collective social institutions rather than the individual-level perceptions that underpin everyday social interactions. Here, we use open, extensible methods to unite diverse theoretical ideas into a common measurement tool for individual prestige. Participants evaluated the perceived prestige of regional variations in accented speech using a pool of candidate scale items generated from free-listing tasks and a review of published scales. Through exploratory and confirmatory factor analyses, we find that our resulting 7-item scale, composed of dimensions we term position, reputation, and information ("PRI"), exhibits good model fit, scale validity, and scale reliability. The PRI scale of individual prestige contributes to the integration of existing lines of theory on the concept of prestige, and the scale's application in Western samples and its extensibility to other cultural contexts serves as a foundation for new theoretical and experimental trajectories across the social and behavioral sciences.

## Introduction

Prestige is a key concept for many disciplines in the social and behavioral sciences, including psychology [1], sociology [2], anthropology [3], and economics [4]. Through its influence on the cultural transmission of knowledge and the dynamics that shape cultural diversity, prestige has been implicated as a crucial component in the evolution of our highly social species [5–8]. These cultural evolutionary dynamics ultimately arise from social interactions between individuals at the microevolutionary level. Therefore, we can consider the individual as the unit that acquires, holds, and benefits from prestige in day-to-day life. Despite the theoretical and practical importance of the prestige concept, few tools have been developed that measure individual prestige, and theoretical and methodological issues may have impeded the efficacy of existing scales.

project (https://osf.io/72v3f/; doi:10.17605/OSF.IO/72V3F).

**Funding:** This work was supported by a grant from the Max Planck Institute for the Science of Human History (http://www.shh.mpg.de/) to MCG and FMJ. The funders provided advisement for the work but had no role in study design, data collection and analysis, decision to publish, or preparation of the manuscript.

**Competing interests:** The authors have declared that no competing interests exist.

A scale of individual prestige that is theoretically and practically meaningful must have validity (e.g. it measures what it is intended to measure) and reliability (e.g. it is consistent in those measurements). When quantifying prestige, the scale must measure perceptions of the traits that constitute prestige and the relative influence these traits have on the general prestige construct. The scale should also assist researchers in accounting for differences in perceptions between groups of respondents—by culture, demographics, or otherwise—in order to avoid being misled by results from inappropriately aggregating across these groups [9–11]. In addition, the scale should be developed using replicable methods to allow for adaptations for use with new groups that may hold different values. Lastly, in developing the scale, researchers should endeavor to be data-driven in their approach and minimally reliant on *a priori* theoretical assumptions [12,13] to reduce the potential bias posed by researchers' expectations and to maximize the real-world utility and validity of the scale. Specifically, participants' responses should be allowed to determine the structure of the scale, rather than fitting a predetermined theoretical model without adequate attention to goodness of fit. Though all research holds inherent biases from its foundations in prior theory and from the implicit biases of the people performing it, these biases can be avoided to a degree by allowing the data itself and the voices of participants to guide the scale development process wherever possible.

Rather than individual prestige, many existing prestige scales focus on the prestige of collective social institutions or constructs, such as organizational prestige (regard for an institution, e.g. [14,15]), brand prestige (status associated with products, e.g. [16,17]), and occupational prestige (standing of professions, e.g. [18–20]), that are not directly derived from or attributable to individual-level traits. Some of the most widely-used "scales" of occupational prestige—including the NORC Duncan Socioeconomic Index [18], the Nakao-Treas Prestige Score [19], and the International Socio-Economic Index of Occupational Status [20] (and its predecessors, e.g. [21])—are not measurement tools, but rather lists of prior composite ratings for each occupation. Researchers obtained some of these existing prestige "scales" (and others, e.g. [22,23]) by directly asking participants to rank others by their own internal concept of prestige, left undefined, or by how participants think society in general would or should rank them. These ambiguities in previous indices of prestige leave findings open to theoretically-biased interpretations [24–26].

The distinction between data-driven and theory-driven research is also relevant when considering the suitability of another published scale for measuring individual prestige: the prestige-dominance scale developed by Cheng et al. [27]. This scale was built to conform to a specific theoretical framework [28] and contrasts "prestige" and "dominance" as opposing unidimensional constructs. To maintain theoretical soundness, Cheng and colleagues chose to retain multiple scale items that did not meet their stated inclusion criteria and contributed to a poorly-fitting final model (CFI < 0.95, GFI < 0.90, RMSEA > 0.05) [27]. Here, for the purpose of developing an accurate measurement tool, we consider that the characteristics of an individual that may contribute to prestige could also overlap with those that contribute to dominance, rather than belonging to either of two fully discrete avenues to status. Previous research [29–32] suggests that peoples' mental models for one or both of these constructs may also be multi-dimensional rather than unidimensional. Importantly, these hypotheses can be assessed using an empirical, theory-neutral approach.

The purpose of our work is to construct a valid and reliable scale of individual prestige, as defined by participants within two broadly "Western" societies—the United States and the United Kingdom—using replicable methods that we intend to be extensible to other contexts and cultures. We take a minimal theoretical approach to prestige, elements of which have been suggested in disparate parts of the literature but never explored together in one measurement tool. Our approach makes three fundamental assumptions about prestige:

1. Prestige can be seen as a trait possessed and used by an individual in the course of everyday social life, distinct from but not independent of the prestige accorded to the societal institutions and constructs of which they may be a part [2,25,33];

2. Prestige is based upon the subjective assessments of others, through the lens of their individually, socially, and culturally acquired beliefs, values, attitudes, and experiences [2,3,25,29,34,35]; and

3. Prestige may be composed of multiple dimensions [2,29–32,36,37], each representing differential contributions from individual, social, or cultural domains.

These do not constitute an exhaustive list of the assumptions involved in the research process. However, we intentionally made no further assumptions about what constitutes prestige or about its specific societal mechanisms and consequences, as our goal was to obtain the necessary information from respondents' own views of prestige in the real world [25]. Our approach was driven to a large degree by the responses of participants, rather than relying on a specific theoretical prestige concept.

One methodological challenge of our approach involved finding a valid, widely recognized signal of prestige that could be presented to participants to evaluate the pool of prospective prestige scale items. Ideally, this instrument would also avoid pre-defining for participants what prestige means. For this purpose, and because this is one component of a larger study on prestige and the transmission of spoken information, we chose to use accented regional variation in speech to highlight differences in individual prestige. Work by sociolinguists has consistently shown that linguistic characteristics such as dialect and accent can index macro-social categories related to prestige (such as class) in the perceptions of listeners, as well as acquiring socially significant meanings of their own. Accents and regional varieties are therefore perceived as strong indicators of prestige and tend to be stable over time [38–41]. Accents are hard-to-fake signals [42] and because accents that are regarded as locally "standard" or associated with desirable upper class membership tend to be evaluated highly by a majority of listeners, they often serve as an index of membership in a high-status group [38,43,44]. Naturally, some disagreement will exist between different demographic groups on the evaluation of particular accents [38,45]. However, the process of developing a measurement scale involves examining the correlations between items to determine the overall structure of the data and is not sensitive to individual differences in evaluations if the relationships between items are consistent. Therefore, our focus here is not on how respondents rate specific accents; in a separate study, we examine the ratings of particular accents in the context of sociolinguistics and cultural evolution [46].

The development of a valid and reliable scale will enable researchers from diverse disciplinary backgrounds to measure individual prestige using a shared prestige concept. The scale can thus contribute to the evaluation and reconciliation of competing theories on prestige and serve as a foundation for the development of new theoretical and experimental trajectories across the social and behavioral sciences.

## Methods and results

The scale development process involved first constructing the prospective scale by collecting items and determining their structure through exploratory factor analysis, then evaluating the fit of the model using confirmatory factor analysis with a separate data set, and finally assessing the validity and reliability of the scale using qualitative and quantitative criteria. We give an expanded description of each step in the process with greater detail on the methods used in S1 Appendix, and tables or figures with an "S1" prefix are contained within that appendix.

### Ethics statement

We obtained written prior informed consent from all participants in this research. Participants that completed surveys through the Amazon Mechanical Turk and Prolific platforms were compensated above hourly minimum wage, in the state of Colorado for US participants and in the UK for participants located there, based upon the time needed to complete the surveys. Participants self-reported demographic information for socioculturally determined constructs such as ethnicity and gender, using categories in accordance with current local and ethical guidelines. Full details on these categories are given in the description of each data set in S1 Metadata.

Prior approval for research protocols was obtained from the Colorado State University Institutional Review Board (protocol #014-16H) and the University of Bristol Faculty of Arts Human Research Ethics Committee (protocols #26561, #31041, and #38323).

### Study 1: Scale construction

We began by conducting a study to generate a pool of words or phrases ("items") related to prestige, reducing the items to those most indicative of prestige, and constructing the scale by establishing the factor structure of those items using exploratory factor analysis ("EFA"). We collected items from three sources: the most salient terms in a free-listing task completed by participants; a previously unpublished pilot study on sociolinguistic prestige; and a review of published scales that measure language attitudes and incorporated a prestige or status dimension. We also collected items from two contrasting domains—"solidarity" and "dynamism"—from published sources [41], to ensure that scale items adequately discriminated between prestige and other unrelated concepts with positive connotations. We did not impose any theoretical structure to the items during data collection: all items were randomized and presented together. We used the resulting 20 items (Table 1) for this study and for the follow-up scale evaluation study.

We recruited participants from the US (*n* = 153) and UK (*n* = 155) to complete an online survey using these items to evaluate the characteristics of four speakers with varying regional accents of English. As a second complementary source of data on perceptions of association between items without involving accents, participants were also asked to group the prestige domain items into like and unlike categories using a triad test [47]. In this exercise, we presented participants with sequences of three items and asked them to eliminate the item that

**Table 1. Pool of attitudinal items retained and used in the scale construction and scale evaluation studies.** Reversed items used in the scale evaluation study are noted parenthetically.

| PRESTIGE | SOLIDARITY | DYNAMISM |
|---|---|---|
| *prestigious* | *friendly* | *aggressive* |
| *wealthy* | *kind (unkind)* | *active* |
| *high social status* | *good-natured* | *confident* |
| *powerful* | *warm* | *enthusiastic* |
| *respected* | *comforting* | |
| *educated* | | |
| *hardworking* | | |
| *successful* | | |
| *intelligent (unintelligent)* | | |
| *reputable* | | |
| *ambitious (unambitious)* | | |

was least like the others, leaving a pair of like items. Participants each completed 55 of these tri-ads, determined by a balanced design in which every possible pair of items appeared exactly three times.

By sequentially applying EFA and eliminating items that failed to reach the predetermined acceptance criteria (see S1 Appendix), we obtained the best-supported factor structure for the full set of attitudinal items across all three domains (Fig 1; **S1.1A Table in** S1 Appendix). This structure supported the division of items into prestige, dynamism, and solidarity domains and the separation of a distinct prestige factor. Using the same criteria, we then determined the best-supported internal factor structure of the attitudinal and triad items in the prestige domain (Fig 2; **S1.1B** and **S1.1C Table in** S1 Appendix). As a result of performing EFA, items within the prestige domain were partitioned into three factors: *wealthy*, *powerful*, and *high social status* in the first factor, hereafter referred to as "position"; *reputable* and *respected* in the second factor, referred to as "reputation"; and *educated* and *intelligent* in the third factor, referred to as "information." We therefore denote the resulting overall factor structure as Posi-tion-Reputation-Information, or "PRI." Subsequent cluster analyses on the same data gener-ated clusters that matched the three PRI factors (**S1.4A Fig in** S1 Appendix), as did results from comparable analyses of the triad data (**S1.4B Fig in** S1 Appendix), supporting the robust-ness of this structure.

## Study 2: Scale evaluation

We then conducted a second study with an independent data set to validate the findings of the scale construction study using confirmatory factor analysis ("CFA"). The validation step evaluates the fit of the structural model proposed by EFA and examines any potential system-atic variance due to sampling [48]. We used the full set of relevant items from the scale

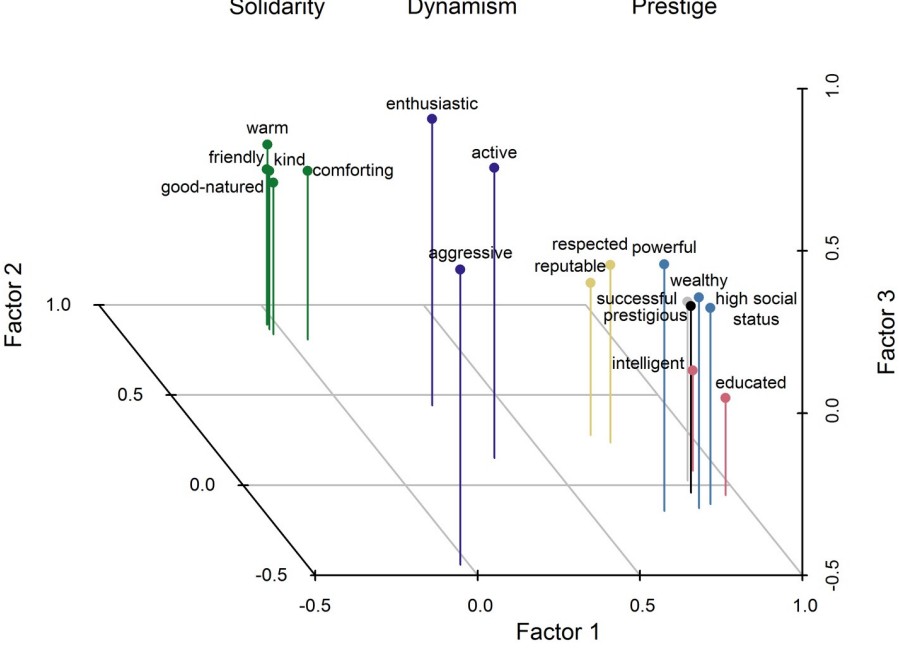

**Fig 1. Overall factor loadings from exploratory factor analysis of attitudinal data.** Visual display of the values in **S1.1A Table in** S1 Appendix. Position, reputation, and information items are shown in light blue, gold, and pink, respectively. Other prestige items are shown in black (*prestigious*, not used in scale) and gray (later dropped from internal prestige structure shown in Fig 2). Solidarity items are in green. Dynamism items are in purple.

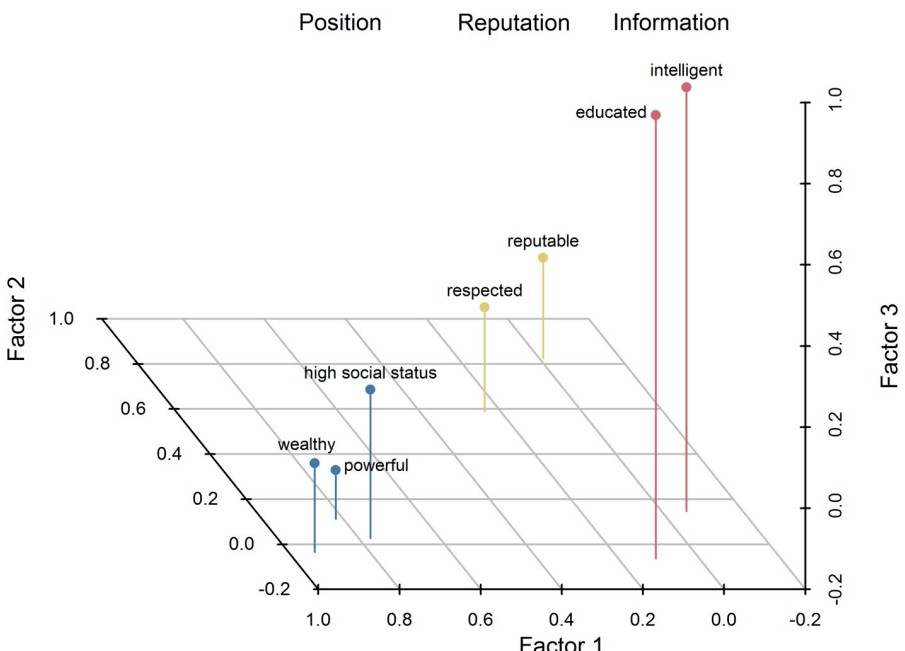

**Fig 2. Prestige domain item loadings from exploratory factor analysis of attitudinal data.** Visual display of the values in **S1.1B Table in** S1 Appendix. Position, reputation, and information items are shown in light blue, gold, and pink, respectively.

construction study in the CFA, with three items presented in reversed form to reduce potential response bias (but this was found to be ineffective, see S1 Appendix).

For this study, we recruited a new, independent sample of participants from the US ($n$ = 151) and UK ($n$ = 144) to provide attitudinal ratings for a greater variety of accented speakers than in the previous study ($n$ = 8 in each country, 4 of which were presented to participants in both countries; see **S1.2 Table in** S1 Appendix), again using an online survey.

After controlling for potential differences between participant demographics, we found that the PRI model exhibited good fit (CFI = 0.959, TLI = 0.983, RMSEA = 0.031 [90% CI: 0.026, 0.036], SRMR = 0.023). Following this validation by CFA, we obtained the complete PRI scale (Fig 3).

## Scale validity and reliability

The PRI scale displayed both validity and reliability in the context of our samples. Using predetermined criteria to judge the acceptability of each index (see S1 Appendix), we found support for the components of construct validity: convergent validity measures exceeded the criterion for all subscales (average variance explained, or "AVE": position = 0.670, reputation = 0.629, information = 0.696) and discriminant validity measures (heterotrait-monotrait ratio, or "HTMT": **S1.5 Table in** S1 Appendix) remained below the threshold in all cases except in one comparison between internal position and information subscales. Reliability measures of internal consistency (coefficients alpha and omega: **S1.6 Table in** S1 Appendix) were high within each PRI subscale ($M$ = 0.813, $SD$ = 0.036) and for the scale as a whole ($M$ = 0.892, $SD$ = 0.018). Criterion validity was demonstrated by high correlations between scale items and a separate *prestigious* item ($M$ = 0.692, $SD$ = 0.097). As added support for the criterion validity of the PRI scale, in a comparative data set the factor scores predicted by the PRI scale were highly correlated with those of the prestige factor of the Cheng et al. [27] prestige-dominance

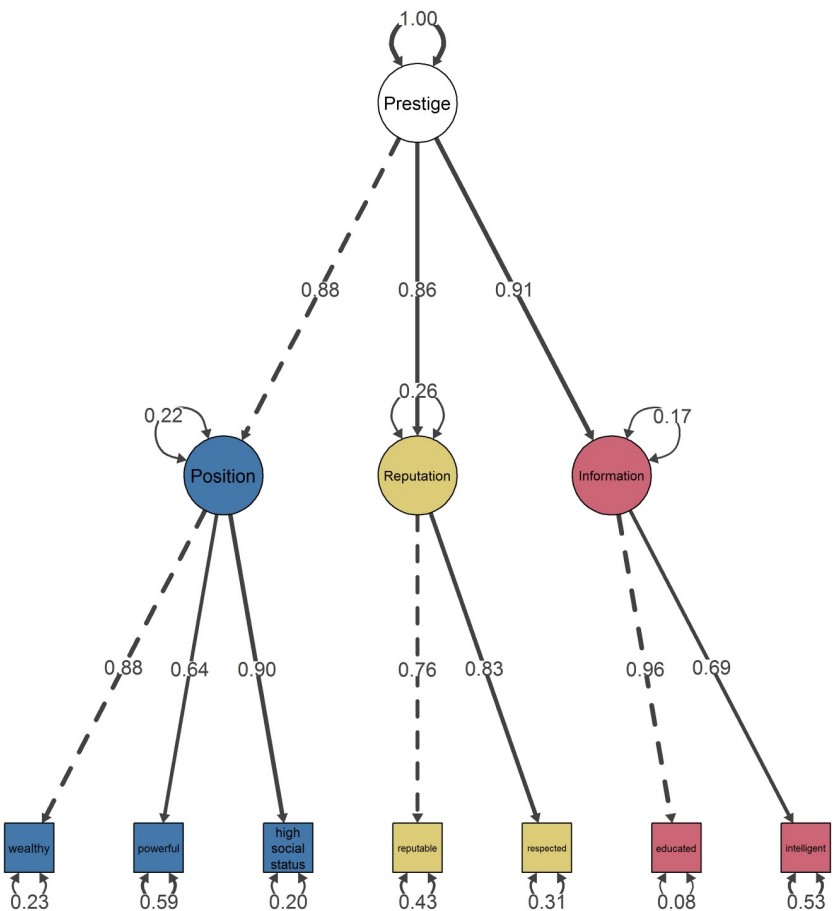

**Fig 3. Path diagram and estimates from confirmatory factor analysis of the Position-Reputation-Information scale model.** Standardized parameter estimates are shown as weighted edges. Residual variances are shown as self-loops. Dotted lines indicate that the loadings of the first indicator of each factor were fixed to 1.0 for estimation.

scale (PRI overall: 0.850, position: 0.805, reputation: 0.861, information: 0.828) and the PRI scale displayed better model fit overall ($\Delta$CFI = 0.025, $\Delta$TLI = 0.029, $\Delta$RMSEA = -0.045, $\Delta$SRMR = -0.064; see S1 Appendix).

These assessments demonstrate that the PRI scale adequately represents the prestige construct and that it is distinct from the other positive traits tested (i.e. solidarity and dynamism). The three subscales (position, reputation, and information) represent cohesive parts of an integrated evaluation of individual prestige while being relatively distinct from one another. Additionally, perceptions of the PRI structure were consistent across respondents and the scale compares well with existing prestige concepts. We take these results as support for the PRI scale as the most accurate and realistic reflection of our participants' internal views on the content and structure of the individual prestige construct.

## Discussion

In the process of developing the PRI scale, we intentionally minimized the role of theory and allowed the structure inherent in the data—structure provided by participants' own internal conceptions of prestige and revealed through factor analysis—to dictate what was most relevant. However, in examining this structure and the constituent items of the scale after its

formation, we found that the PRI prestige construct is highly consistent with different streams of prior research on prestige. The terms chosen to represent the three subscales, "position," "reputation," and "information," characterize three relatively distinct axes of individual prestige, and we examine each in turn.

The position components of the scale signify an individual's relative place in the social hierarchy, determined to a large extent by the circumstances of their birth, family, and inheritance. Max Weber, in his classic theory of social stratification, argued that one's social position can be attributed to three dimensions: economic "class," or wealth; "status," or honor gained through prestige; and "party," or political power and influence [49,50]. We suggest that Weber's three dimensions parallel the three items found in the position subscale (*wealthy*, *high social status*, and *powerful*) and, like Weber's components, the three position items are likely to be interrelated. This finding could be a contingency arising from the set of items used in this study; however, we find the connection to Weber's work to be a useful point of comparison that supports the continuing utility of Weber's ideas in sociological theory and practice [51].

The items in the reputation subscale (*reputable* and *respected*) relate to social opinion and esteem and are terms frequently used to describe prestige (e.g. [14,15,52]), and are even used synonymously with it (e.g. [53]). In the sociological literature on prestige, reputation and respect have the connotation of a collective judgment of character independent of individual variation in judgments [2]. Reputation and respect represent the general societal evaluation of an individual in a certain position or role, subjectively interpreted through social and cultural values. By contrast, the items in the position subscale may be established through privilege without necessarily undergoing the same degree of collective evaluation [49,50].

The third subscale, information, and its items (*educated* and *intelligent*) represent the value placed by society on the holders of wisdom, expertise, and learning. These constructs are supported by findings from the occupational prestige literature showing that—in a stratified society with specialized occupations—an individual's educational background and achievement are highly predictive of their future occupational class which, in turn, contributes significantly to individual prestige (e.g. [54–56]). Occupation as a social construct spans all three subscales; for instance, merely holding a particular occupation can lead to assumptions that the holder is *wealthy*, *respected*, and *intelligent* or, conversely, that they are none of these. However, occupation alone is not sufficient to explain individual prestige, as interactions with other constructs such as race and gender lead to inequalities in prestige and its components [57].

The salience of the information subscale and its focus on information holders could also indicate support for arguments from information theory about the evolution of prestige and its role in cultural transmission. The information theory-based account, presented alongside (but not integral to) the dichotomous prestige-dominance distinction by Henrich & Gil-White [28], asserts that individuals gain prestige by having desirable skills and knowledge that others compete within a social group for the opportunity to learn. Alternatively, an occupation attained through greater education could be another avenue to wealth and power. This question, and to what extent—if any—some form of the information subscale would be relevant to prestige across the diversity of non-Western or non-industrialized societies remains open to future study.

Indeed, there is a great need to explore concepts of prestige cross-culturally to reach beyond the perspectives given by Western and westernized participants. Many existing prestige indices have been explicitly promoted for their universality, in spite of having been developed using data almost exclusively from "WEIRD" (Western, Educated, Industrialized, Rich, and Democratic) societies [58] in the 1960's, '70s, and '80s. The utility of these indices across cultures and over the significant span of time and sociocultural change that has occurred since they were developed has been called into question [9–11,29,59].

The concept of prestige, the individual components that comprise prestige, the degree of importance attached to each component, and the relationships between components are all—to some degree—culturally constructed and malleable through cultural evolutionary processes. Therefore, we recognize that the PRI scale is not universally applicable, as this is an unrealistic expectation. We developed the PRI scale using data collected from adults in the highly WEIRD societies of the United States and United Kingdom and it should not be generalized beyond the WEIRD context without adequate validation. The high degree of consistency in the PRI structure across our representative samples of demographically diverse participants in the US and UK suggests that the PRI scale should function well across other highly Westernized, English-speaking societies. However, distinct demographic or cultural groups within these societies may hold different values and have substantially different internal models of prestige. For these reasons, and in the interest of following best practices in psychometrics [60], we strongly recommend testing the validity and reliability of the PRI scale with each application and testing for invariance across as many demographic variables as may be potentially relevant.

We have made the process of constructing and validating the PRI scale extensible to any additional population for which a scale of individual prestige is needed, through the emphasis on participants in the item generation and evaluation stages, the use of straightforward and appropriate methods and criteria (see S1 Appendix), the use of open-source analytical tools, and the open sharing of all data and code used to run analyses (see S1 Data). A variant of the PRI scale can be constructed by repeating these methods in a new group, with awareness and care for local cultural norms and power structures. Examining systematic differences in responses and extending the PRI scale to other contexts and cultures can further improve the representation and inclusion of minority and non-Western perspectives on prestige, and we argue that this is the most important avenue for future research presented by this study.

The PRI scale for the measurement of individual prestige fills a crucial niche by establishing a measurement tool driven by the real-world perceptions of individuals across two Western societies. The PRI scale enables the study of prestige—a central yet divisive concept throughout the social and behavioral sciences—using a common foundation, which we hope will encourage fruitful engagement, conversation, and collaboration that spans across disciplinary boundaries. We have shown the broad utility of this scale for conducting research by finding support for the PRI structure in both of two separate sources of data: attitudinal responses to variations in accented speech, and triadic conceptual associations absent the sociolinguistic context. Future studies using additional stimuli—such as photos, videos, or character vignettes—can explore the potential of the PRI scale at assessing individual prestige as it is represented across a broad array of experimental contexts.

Future research should endeavor to untangle the complex and varied patterns in how prestige is perceived and how it operates in the practice of real social interactions across the breadth of human experience. The availability of the PRI scale allows researchers to explore in greater detail the relationships between different aspects of prestige, dominance, status, and success. Some of these relationships may be quite complex, or even circular, as suggested by the presence of *high social status* as an indicator of prestige within the position subscale (whereas scholars would normally consider prestige to be a contributor to status) or by the possible contributions of specific indicators like *educated* toward other indicators like *wealthy*. Additionally, there may be some degree of overlap between the construct of prestige, as measured by the PRI scale and the prestige factor of the Cheng et al. [27] prestige-dominance scale, and other related concepts like dominance and leadership. Indeed, prestige and dominance have been found to co-occur within individuals in humans and some non-human animals [61–63]. Thus, many questions remain about the breadth and interconnectedness of the varied

routes to the acquisition of social status. We view the establishment of the PRI scale as a necessary step toward a more integrated and comprehensive understanding of prestige, through the clarification of preceding debates and the beginning of new lines of inquiry into the core concepts that shape interactions, relationships, social structures and inequality, and the evolution of culture.

## Supporting information

**S1 Appendix. Supplementary methods.** Complete description of methods used and additional supporting results, in Word DOCX format.
(DOCX)

**S1 Form. Position-reputation-information scale administration form.** List of final Position-Reputation-Information items with 7-point Likert-type rating scale, in Word DOCX format. Additional items of interest can be included and the order of all items should be randomized prior to presentation. We do not recommend reverse-scoring any items, due to observed biases in responses (see text).
(DOCX)

**S1 Data. Data sets and R code.** Archive of data sets (as RDS and CSV) and R code (pri_analyses, as R Markdown script and PDF document) used for all analyses and the generation of figures, in ZIP format. Data sets included are: free list data (list_f and data_f), pilot study attitudinal data (data_p), scale construction study attitudinal data (data_s) and triad data (data_t), scale evaluation study attitudinal data (data_c), and criterion validity comparative attitudinal data (data_v). The final confirmatory lavaan model object (cfa_pri.RDS) is also included. See S1 Metadata for full descriptions of data sets, types, and variables.
(ZIP)

**S1 Metadata. Data set descriptions.** Complete information on data sets and variables contained in S1 Data, in Word DOCX format.
(DOCX)

**S1 Table. Speaker demographics.** Demographic information for speakers used in scale construction and scale evaluation studies, in Word DOCX format. Contains information on speaker accent, recording ID (if from IDEA, see Acknowledgements), country, state, age (in years) at time of recording, place of birth, place raised (for majority of childhood), gender, ethnicity, occupation, education, other places lived, other possible influences on speech, identity of recorder, and recording date.
(DOCX)

## Acknowledgments

We thank Russell Gray for his advisement of this project, Jerry Vaske and Jeffrey Snodgrass for their valuable feedback, Mark Prince for help with structural equation modeling methods, and Cory Holland for advice on speech recording and accent evaluation. We also thank all of our participants for their important contributions to this work. We express our gratitude to Julie Gros-Louis and two anonymous reviewers for their contributions to evaluating and improving this paper.

The recordings used in this project—except for American West (Urban) and Wales, recorded by the authors—are used by special permission of the International Dialects of English Archive, online at http://www.dialectsarchive.com, and we are grateful to IDEA for the licensed use of these recordings. *Comma Gets a Cure* is copyright 2000 Douglas N. Honorof,

Jill McCullough & Barbara Somerville, text available online at: http://www.dialectsarchive.com/comma-gets-a-cure.

The color palettes used in figures are derived from a technical note by Paul Tol (available at: https://personal.sron.nl/~pault/data/colourschemes.pdf) and are optimized for color-blind readers.

## Author Contributions

**Conceptualization:** Alarna N. Samarasinghe, Fiona M. Jordan, Michael C. Gavin.

**Data curation:** Richard E. W. Berl, Alarna N. Samarasinghe.

**Formal analysis:** Richard E. W. Berl.

**Funding acquisition:** Fiona M. Jordan, Michael C. Gavin.

**Investigation:** Richard E. W. Berl, Alarna N. Samarasinghe.

**Methodology:** Richard E. W. Berl, Alarna N. Samarasinghe, Fiona M. Jordan, Michael C. Gavin.

**Project administration:** Richard E. W. Berl, Fiona M. Jordan, Michael C. Gavin.

**Resources:** Richard E. W. Berl, Fiona M. Jordan, Michael C. Gavin.

**Software:** Richard E. W. Berl.

**Supervision:** Fiona M. Jordan, Michael C. Gavin.

**Validation:** Richard E. W. Berl.

**Visualization:** Richard E. W. Berl.

**Writing – original draft:** Richard E. W. Berl.

**Writing – review & editing:** Richard E. W. Berl, Alarna N. Samarasinghe, Fiona M. Jordan, Michael C. Gavin.

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
