## [Decision Letter · Decision Letter 0]

25 Feb 2020

PONE-D-19-30994

The Position-Reputation-Information (PRI) scale of individual prestige

PLOS ONE

Dear Dr. Berl,

Thank you for submitting your manuscript to PLOS ONE. After careful consideration, we feel that it has merit but does not fully meet PLOS ONE’s publication criteria as it currently stands. Therefore, we invite you to submit a revised version of the manuscript that addresses the points raised during the review process.

Both of the reviewers suggest minor revisions. Please address all of the comments and suggestions in your revision. 

We would appreciate receiving your revised manuscript by Apr 10 2020 11:59PM. To enhance the reproducibility of your results, we recommend that if applicable you deposit your laboratory protocols in protocols.io, where a protocol can be assigned its own identifier (DOI) such that it can be cited independently in the future. For instructions see: http://journals.plos.org/plosone/s/submission-guidelines#loc-laboratory-protocols

We look forward to receiving your revised manuscript.

Kind regards,

Julie Jeannette Gros-Louis, PhD

Academic Editor

PLOS ONE

Journal Requirements:

Reviewers' comments:

Reviewer's Responses to Questions

**Comments to the Author**

1. Is the manuscript technically sound, and do the data support the conclusions?

Reviewer #1: Yes

Reviewer #2: Yes

2. Has the statistical analysis been performed appropriately and rigorously? 

Reviewer #1: Yes

Reviewer #2: Yes

3. Have the authors made all data underlying the findings in their manuscript fully available?

Reviewer #1: Yes

Reviewer #2: Yes

4. Is the manuscript presented in an intelligible fashion and written in standard English?

Reviewer #1: Yes

Reviewer #2: Yes

5. Review Comments to the Author

Reviewer #1: The authors generate a prestige scale that, in their view, avoids theoretical assumptions that limit the applicability of past scales. As such, I agree that this is a valuable contribution to the literature. I have a number of comments and questions for the authors’ consideration- see below. In particular, the independence of the scale development from any particular theory doesn’t mean there aren’t assumptions that guide the author’s choices.

Line 45: Joey Cheng and colleagues would take issue with this claim. While you provide substantive critique of their scale later, the claim here is over-stated.

Line 56: describe more what you mean by theory neutral. I think you are arguing, per your critiques of Cheng et al.’s scale, that the prestige concept should be operationalized independent of theories of its origin/evolution or application. But whenever anything is operationalized, there is theory at least implicitly guiding the process, in terms of how the world should be categorized.

Lines 94-103: these assumptions can encompass perhaps any trait. I would think at a minimum there would have to be some greater delimiting of how you gather respondents conceptions of prestige, which incorporates some modicum of theory. For example, that not all traits we observe in others are evaluated equally (some traits seen as more favorable or rewarding than others), which affects how people socialize with each other, in particular who receives more deference than others. Indeed, you describe use of accents in this study because of their index of membership in high-status groups (line 123).

Lines 125-126: the following requires more explication: “our focus is not on how respondents rate specific accents but on the relationships between the items used in the evaluation of prestige.”

Line 159: based on the extreme generality of the three assumptions, its not clear how you could reduce the items in the way you did. What criteria was used to evaluate those most “indicative of prestige”? Relatedly, how did you define the most “salient terms” (line 162) in the free-listing task?

Line 166-167: why should items associated with solidarity and dynamism necessarily be distinct from items people associate with prestige, let alone theoretical conceptions of prestige? Isn’t this imposing theory contrary to earlier claims? It may be the case, for example, that people more weakly associate intelligence with their notion of prestige if intelligence (information) isn’t also correlated with confidence (dynamism) or friendliness (solidarity).

Line 179: the triad test requires explication

Lines 182-184: before describing EFA within the prestige domain, make it clearer in the text that the EFA applied to all items (irrespective of domain) provided evidence of a prestige factor distinct from dynamism and solidarity.

Lines 265-272: I’m not convinced that the position factor is necessarily cleaved into Weber’s categories. While the 3 items in the factor mirror Weber’s categories, this may owe a lot to idiosyncrasies in item selection prior to EFA and CFA. Furthermore, Weber’s categories don’t necessarily carve nature at its joints, like most social science typologies. Be more circumspect in your comparison here.

Line 284: the occupational prestige literature is as reflective of the position and reputation domains as the information domain. Compared to janitors, doctors earn more money, are esteemed, and have more knowledge, for example. And may be the case that occupational prestige is driven more by the position domain or reputation domain than the information domain per se.

Pages 17-20: discussion here on cross-cultural considerations and interactions among prestige domains is good and anticipated many other comments I had. I would cite the following that argues that dominance-prestige distinction does not comport with their tendency to co-occur within the same individuals, whether in human or non-human societies: https://www.ncbi.nlm.nih.gov/pubmed/25947621. Your prestige scale may be tapping such convergence of status-generating attributes.

Reviewer #2: Manuscript PONE-D-19-30994 describes the data-driven development of a new scale for measuring dimensions of prestige. My first impression was that this scale did not contribute anything new, given the popularity of the Cheng et al. dominance-prestige measure. However, upon closer examination of the authors’ arguments for the psychometric inadequacy of the dominance-prestige scale, I was persuaded that the introduction of a less theoretically-biased scale into the literature may help to advance research on human status. Moreover, the methodological rigor with which the PRI scale was developed suggests that this new scale is more psychometrically sound than currently popular alternative provides, and the methods provide a good model for future scale development. The authors are also refreshingly careful not to generalize beyond the populations examined in their study, while providing a useful and concrete framework for extending the PRI scale cross-culturally. Although there are some areas that need to be improved with minor revisions, this paper and scale should be published. I outline some suggestions for improvement and revision below.

I generally agree that it is a plus that the authors avoided pre-defining prestige for participants. But it seems that the authors’ choice to use voice recordings as the stimuli is not well defended. Why not use photos, character vignettes, videos, or some other stimuli that is less loosely tied to individual differences in prestige than regional accents? Some discussion of how this choice may have altered the manifest dimensions of prestige is warranted.

There is a lot of important information about scale development and validity that has been relegated to the supplementary material. As a result, the scale validity and reliability section of the main text feels relatively weak (although the evidence for validity and reliability is strong). Many of the intercorrelations between existing measures may be important to readers and researchers who are attempting to evaluate the usefulness of the PRI in their own research and should be made clearly available in the main text. I’d recommend moving as much information from the “Scale validity and reliability” in the appendix to the corresponding section in the main text.

The paper is generally very clearly written and easily understandable, but there are several grammatical errors and possible typos that could be addressed with another round of careful revisions.

6. PLOS authors have the option to publish the peer review history of their article (what does this mean?). If published, this will include your full peer review and any attached files.

Reviewer #1: No

Reviewer #2: No

---

## [Author Response · Author response to Decision Letter 0]

19 May 2020

Copied from Response to Reviewers document:

Reviewer #1

The authors generate a prestige scale that, in their view, avoids theoretical assumptions that limit the applicability of past scales. As such, I agree that this is a valuable contribution to the literature. I have a number of comments and questions for the authors’ consideration- see below. In particular, the independence of the scale development from any particular theory doesn’t mean there aren’t assumptions that guide the author’s choices.

We appreciate this point, and we have made revisions in response to the comments below to acknowledge that our approach and our operationalization of the prestige concept is not free from theoretical assumptions.

Line 45: Joey Cheng and colleagues would take issue with this claim. While you provide substantive critique of their scale later, the claim here is over-stated.

We have toned down the language here (lines 44-46) to specify that other scales do exist but may be hampered by theoretical and methodological concerns, which are explained further in subsequent paragraphs.

Line 56: describe more what you mean by theory neutral. I think you are arguing, per your critiques of Cheng et al.’s scale, that the prestige concept should be operationalized independent of theories of its origin/evolution or application. But whenever anything is operationalized, there is theory at least implicitly guiding the process, in terms of how the world should be categorized.

We agree with the reviewer and recognize our error in framing here. We have added additional detail (lines 57-65) to explain that we made an effort to be minimally reliant on theory but not free of all theoretical assumptions, as a way to reduce the bias inherent in the process.

Lines 94-103: these assumptions can encompass perhaps any trait. I would think at a minimum there would have to be some greater delimiting of how you gather respondents conceptions of prestige, which incorporates some modicum of theory. For example, that not all traits we observe in others are evaluated equally (some traits seen as more favorable or rewarding than others), which affects how people socialize with each other, in particular who receives more deference than others. Indeed, you describe use of accents in this study because of their index of membership in high-status groups (line 123).

We added wording to this section (lines 111-116) to note that these are our fundamental assumptions and not an exhaustive list. The additional explanation given around the role of theory, above, should also help address this concern.

Lines 125-126: the following requires more explication: “our focus is not on how respondents rate specific accents but on the relationships between the items used in the evaluation of prestige.”

We have added additional information here (lines 132-138) to explain that the focus of scale construction methods is on determining overall structure through relationships between items, rather than evaluating individual ratings.

Line 159: based on the extreme generality of the three assumptions, its not clear how you could reduce the items in the way you did. What criteria was used to evaluate those most “indicative of prestige”? Relatedly, how did you define the most “salient terms” (line 162) in the free-listing task?

This sentence is intended to serve as an overview of the methods used, and each is explained in more detail in the following paragraphs and especially in the supplementary appendix. To answer the reviewer’s questions: Items were determined to be indicative of prestige if they had partitioned together with the prestigious item in the exploratory factor analyses and cluster analyses of participants’ responses in Study 1, and had adequate fit within that factor (criteria as listed in appendix: primary factor loading with absolute value > 0.32; cross-loadings with absolute values < 0.32; gap between primary and cross-loadings > 0.2; communality > 0.4; from Costello & Osborne 2005). Salience values for items were calculated using Smith’s S and arranged in a scree plot, and a cutoff point was chosen as recommended by Bernard (2011). This information is available in the appendix, in which we included a comprehensive amount of detail for the purpose of facilitating replication. We intend that answers to any specific questions on methods by readers will be found there.

Line 166-167: why should items associated with solidarity and dynamism necessarily be distinct from items people associate with prestige, let alone theoretical conceptions of prestige? Isn’t this imposing theory contrary to earlier claims? It may be the case, for example, that people more weakly associate intelligence with their notion of prestige if intelligence (information) isn’t also correlated with confidence (dynamism) or friendliness (solidarity).

Though the desire to distinguish prestige from solidarity and dynamism comes from theory, these domain separations were not imposed on our results. Rather, the structure predicted by theory emerged from the analysis of the data and we interpreted it as supportive of prior theory. We added a note to the manuscript clarifying that we did not impose structure on the items (lines 179-181), and the edit below in response to lines 182-184 also helps to clarify this point. It is possible that some items displayed relationships similar to the one described by the reviewer, but these items would have been eliminated for having high cross-loadings (i.e. by being strongly associated with more than one factor/domain) and thus are not present in the final scale, which sought only the most reliable single-factor indicators.

Line 179: the triad test requires explication

Additional detail was added to this section (lines 193-196) to explain the process of a triad test, and more is available in the appendix.

Lines 182-184: before describing EFA within the prestige domain, make it clearer in the text that the EFA applied to all items (irrespective of domain) provided evidence of a prestige factor distinct from dynamism and solidarity.

This is a very useful addition, and we have added text (lines 199-202) to note that the three-domain structure of prestige, dynamism, and solidarity was supported by the results and that the prestige factor is distinct. This also helps to address the concern above about the theory behind the domains.

Lines 265-272: I’m not convinced that the position factor is necessarily cleaved into Weber’s categories. While the 3 items in the factor mirror Weber’s categories, this may owe a lot to idiosyncrasies in item selection prior to EFA and CFA. Furthermore, Weber’s categories don’t necessarily carve nature at its joints, like most social science typologies. Be more circumspect in your comparison here.

We appreciate this criticism and changed our language (lines 290-295) to suggest that there may be links between our findings and Weber’s categories, with appropriate caution in interpretation due to the potential of it being a contingency that arose from our list of items.

Line 284: the occupational prestige literature is as reflective of the position and reputation domains as the information domain. Compared to janitors, doctors earn more money, are esteemed, and have more knowledge, for example. And may be the case that occupational prestige is driven more by the position domain or reputation domain than the information domain per se.

We agree with the reviewer and did not mean to suggest that the occupational prestige literature does not speak to the other two domains as well. We added text here (lines 310-315) to clarify that occupational prestige has effects across all three subscales, but that it is an imperfect measure of individual prestige.

Pages 17-20: discussion here on cross-cultural considerations and interactions among prestige domains is good and anticipated many other comments I had. I would cite the following that argues that dominance-prestige distinction does not comport with their tendency to co-occur within the same individuals, whether in human or non-human societies: https://www.ncbi.nlm.nih.gov/pubmed/25947621. Your prestige scale may be tapping such convergence of status-generating attributes.

As recommended, we have added this point and citation to the discussion (lines 379-381). We thank the reviewer for their many thoughtful and valuable contributions to this manuscript.

Reviewer #2

Manuscript PONE-D-19-30994 describes the data-driven development of a new scale for measuring dimensions of prestige. My first impression was that this scale did not contribute anything new, given the popularity of the Cheng et al. dominance-prestige measure. However, upon closer examination of the authors’ arguments for the psychometric inadequacy of the dominance-prestige scale, I was persuaded that the introduction of a less theoretically-biased scale into the literature may help to advance research on human status. Moreover, the methodological rigor with which the PRI scale was developed suggests that this new scale is more psychometrically sound than currently popular alternative provides, and the methods provide a good model for future scale development. The authors are also refreshingly careful not to generalize beyond the populations examined in their study, while providing a useful and concrete framework for extending the PRI scale cross-culturally. Although there are some areas that need to be improved with minor revisions, this paper and scale should be published. I outline some suggestions for improvement and revision below.

We greatly appreciate the reviewer’s willingness to fairly examine the manuscript and their support for its improvement.

I generally agree that it is a plus that the authors avoided pre-defining prestige for participants. But it seems that the authors’ choice to use voice recordings as the stimuli is not well defended. Why not use photos, character vignettes, videos, or some other stimuli that is less loosely tied to individual differences in prestige than regional accents? Some discussion of how this choice may have altered the manifest dimensions of prestige is warranted.

We note in the manuscript (lines 120-123) that this study is part of a larger project to examine the effects of cultural transmission biases (including prestige) on spoken information and therefore we needed to develop a scale that was appropriate for that context. However, we find support for the broader application of the scale beyond vocal stimuli in our results from the (written) triad tests, which yielded the same factor and cluster structures as the vocal ratings. Notably, other stimuli such as photos or videos, though commonly used, present their own sets of challenges, limitations, and inherent biases. We have added a note to the discussion (lines 364-367) that future research should explore applications using alternative stimuli that the reviewer suggests.

There is a lot of important information about scale development and validity that has been relegated to the supplementary material. As a result, the scale validity and reliability section of the main text feels relatively weak (although the evidence for validity and reliability is strong). Many of the intercorrelations between existing measures may be important to readers and researchers who are attempting to evaluate the usefulness of the PRI in their own research and should be made clearly available in the main text. I’d recommend moving as much information from the “Scale validity and reliability” in the appendix to the corresponding section in the main text.

These sections were moved to the supplementary material based on prior PLOS ONE editorial comments that the manuscript was too lengthy. The current version reflects a general overview of the methods and results, with references to the appendix. We see the merit of both approaches, given the length and depth of detail included in the methods. If the current editor agrees that the content from the appendix should be moved back to the main body of the manuscript, we are happy to do so.

The paper is generally very clearly written and easily understandable, but there are several grammatical errors and possible typos that could be addressed with another round of careful revisions.

We were surprised to learn this after having done several internal checks and appreciate the reviewer’s attention to these issues. We have carefully combed through the manuscript and corrected the errors we could find. If the reviewer knows of any remaining instances, we would appreciate being pointed to them.

---

## [Editor Report · Decision Letter 1]

27 May 2020

The Position-Reputation-Information (PRI) scale of individual prestige

PONE-D-19-30994R1

Dear Dr. Berl,

We are pleased to inform you that your manuscript has been judged scientifically suitable for publication and will be formally accepted for publication once it complies with all outstanding technical requirements.

With kind regards,

Julie Jeannette Gros-Louis, PhD

Academic Editor

PLOS ONE
---

## [Editor Report · Acceptance letter]

16 Jun 2020

PONE-D-19-30994R1 

The Position-Reputation-Information (PRI) scale of individual prestige 

Dear Dr. Berl:

I'm pleased to inform you that your manuscript has been deemed suitable for publication in PLOS ONE. Congratulations! Your manuscript is now with our production department. 

Kind regards, 

on behalf of

Dr. Julie Jeannette Gros-Louis 

Academic Editor

PLOS ONE